# Recent Advances in Adult Post-Transplant Lymphoproliferative Disorder

**DOI:** 10.3390/cancers14235949

**Published:** 2022-12-01

**Authors:** Mariam Markouli, Fauzia Ullah, Najiullah Omar, Anna Apostolopoulou, Puneet Dhillon, Panagiotis Diamantopoulos, Joshua Dower, Carmelo Gurnari, Sairah Ahmed, Danai Dima

**Affiliations:** 1Department of Internal Medicine, Laikon General Hospital, National and Kapodistrian University of Athens, 11527 Athens, Greece; 2Department of Translational Hematology and Oncology Research, Cleveland Clinic Foundation, Cleveland, OH 44195, USA; 3Division of Infectious Diseases, Massachusetts General Hospital, Harvard Medical School, Boston, MA 02114, USA; 4Department of Internal Medicine, Cleveland Clinic Foundation, Cleveland, OH 44195, USA; 5Department of Hematology and Medical Oncology, Tufts Medical Center, Boston, MA 02111, USA; 6Department of Lymphoma-Myeloma, University of Texas MD Anderson Cancer Center, Houston, TX 77030, USA; 7Department of Hematology and Medical Oncology, Taussig Cancer Institute, Cleveland Clinic Foundation, Cleveland Clinic, Cleveland, OH 44195, USA

**Keywords:** post-transplant lymphoproliferative disorder, Epstein–Barr virus, immunotherapy, rituximab, transplantation

## Abstract

**Simple Summary:**

Post-transplant lymphoproliferative disorder (PTLD) is a potentially life-threatening complication of mainly solid organ and, less frequently, allogeneic hematopoietic stem-cell transplantation, with a reported incidence of 2 to 20%. PTLD has a complex pathogenesis and not all aspects are well understood to date; however, a proportion of cases are strongly related to Epstein–Barr virus. Therapy mainly depends on the histologic subtype; however, the heterogeneity of the disease and lack of clinical trials create gaps in evidence-based management of these patients. In this review, we discuss the pathogenesis, classification, and risk factors of PTLD. We further analyze common treatment strategies and describe the latest advances in disease management and prevention, including novel therapies.

**Abstract:**

PTLD is a rare but severe complication of hematopoietic or solid organ transplant recipients, with variable incidence and timing of occurrence depending on different patient-, therapy-, and transplant-related factors. The pathogenesis of PTLD is complex, with most cases of early PLTD having a strong association with Epstein–Barr virus (EBV) infection and the iatrogenic, immunosuppression-related decrease in T-cell immune surveillance. Without appropriate T-cell response, EBV-infected B cells persist and proliferate, resulting in malignant transformation. Classification is based on the histologic subtype and ranges from nondestructive hyperplasias to monoclonal aggressive lymphomas, with the most common subtype being diffuse large B-cell lymphoma-like PTLD. Management focuses on prevention of PTLD development, as well as therapy for active disease. Treatment is largely based on the histologic subtype. However, given lack of clinical trials providing evidence-based data on PLTD therapy-related outcomes, there are no specific management guidelines. In this review, we discuss the pathogenesis, histologic classification, and risk factors of PTLD. We further focus on common preventive and frontline treatment modalities, as well as describe the application of novel therapies for PLTD and elaborate on potential challenges in therapy.

## 1. Introduction

Post-transplant lymphoproliferative disorder (PTLD) is the most common malignancy in recipients of solid organ transplantation (SOT), excluding nonmelanoma cutaneous and in situ cervical malignancies [1]. In contrast, PTLD occurs in a minority of patients following allogeneic hematopoietic stem-cell transplantation (alloHSCT) [2,3]. Regardless of the setting, PTLD has been associated with significant morbidity and mortality. PTLD mainly arises from B cells but may rarely develop from T or NK cells, which account for 2–15% of all PTLD cases. More than half of B-cell PTLDs are driven by abnormal expansion of Epstein–Barr virus (EBV)-infected B cells. Non-B-cell PTLDs are usually not EBV-related, occur after a long latency period post-transplantation, and are associated with a more aggressive course and poor prognosis, with a median overall survival (OS) of ~6 months [4]. Higher PTLD rates have been observed in multiorgan and intestinal (<20%), lung (3–10%), and heart (2–8%) transplants due to the higher level and prolonged need of immunosuppressive therapy (IST) to prevent allograft rejection [1,5,6].

The primary management of PTLD includes reduction or complete cessation of IST. However, this is not always possible considering the risk for allograft loss or dysfunction, especially in cases of vital organ transplantation, such as heart transplant. In this context, a variety of therapeutic options have been proposed including chemo- and immunotherapy, radiation therapy, and various novel treatments, all with variable results [7]. While there are no guidelines for treatment, the type of lesion and EBV status generally drives the type of therapy based on consensus statements [8,9,10]. Further research is necessary to establish concrete treatment recommendations. In this review, we discuss the clinical presentation, pathophysiology, and risk factors of PTLD. We further analyze common treatment strategies, and describe the latest advances in disease management and prevention.

## 2. Pathogenesis

The pathogenesis of PTLD is complex and not fully understood. The dominant theory is that the required post-transplant IST negatively impacts the ability of cytotoxic T cells for immune eradication via multiple mechanisms, including inability to produce vital cytokines for immune destruction, such as interferon gamma (IFN-γ), interleukin 2 (IL-2), and tumor necrosis factor-alpha (TNF-α). This dysfunction allows abnormal clones of B lymphocytes (most commonly infected with EBV) to proliferate, ultimately transforming to PTLD.

In most instances, PLTD has been strongly associated with EBV infection [11,12]. EBV can be transmitted from a seropositive donor to a previously EBV-naïve/seronegative recipient and manifest as a primary infection, but PTLD can also be a result of EBV reactivation in a seropositive recipient who had previously acquired EBV via environmental exposure, in the setting of immunosuppression [13]. The virus remains latent in B lymphocytes or progresses with viral replication and B-cell lysis. Primary EBV infections in immunocompetent adults are usually self-limited and do not result in any significant complications because proliferation of B cells is normally suppressed by a virus-specific cytotoxic T-lymphocyte response, eradicating the majority of phenotypically abnormal B cells infected with EBV [13]. Disruption in T-cell surveillance, in the setting of post-transplant iatrogenic immunosuppression, can lead to unchecked B-cell proliferation and transformation, hence developing into PTLD [13].

Less is known about the pathogenesis of EBV-negative PTLD. It is believed to be similar to that of de novo non-Hodgkin's lymphoma (NHL) in immunocompetent hosts based on available molecular and immunohistochemical data, rather than true PTLD [14,15]. Hit-and-run EBV infections, indicating an EBV infection that initiates the pathogenesis of PTLD and then resolves, infections with other herpesviruses (e.g., cytomegalovirus (CMV), human herpesvirus 6 (HHV-6) or other viruses), persistent antigen stimulation by the graft, and long-term IST, have all been proposed as potential mechanisms [13,16]. From a clinical perspective, EBV-negative PTLD tends to occur in older transplant recipients, has a longer latency after transplant, is typically associated with SOT, and tends to have high-risk cytogenetic features [17].

EBV-positive PTLD has a different molecular profile and tumor microenvironment changes compared to EBV-negative PTLD [18,19,20,21]. Commonly encountered chromosomal aberrations are gains of 7q, 11q24–q25, and del(4)(q25–q35); combinations of the aberrations tend to occur more frequently in PTLD-negative cases, as they typically harbor more complex cytogenetics compared to EBV-positive PTLD [12]. A distinctive chromosomal abnormality in EBV-positive PTLD is the gain of 9p21, which causes changes in cyclin-dependent kinase inhibitor 2A (*CDKN2A*) expression and thus changes in cell cycle regulation [12]. Gain/amplification of 9p21 has also been found to lead to upregulation of the immune checkpoint receptor ligand, programmed cell death 1 ligand 2 (PD-L2), in EBV-positive cases that is believed to help escape immune surveillance [15]. In contrast, gain of 3/3q encoding the transcription factor Forkhead box p1 (*FOXP1*) is encountered in EBV-negative PTLD. This leads to enhanced expression of *FOXP1*, which appears to be critically related to the pathogenesis of EBV-negative PTLD. *TP53* mutations also appear to be more common in EBV-negative cases [12]. 

EBV has also been found to cause changes in the expression of *BCL6* and *MYC*, as well as activation of the NF-kB, PI3K/AKT/mTOR, and BCL2 molecular pathways, ultimately resulting in the malignant PTLD phenotype [22]. On an epigenetic level, EBV can alter the microRNA expression [23,24], which variably impacts gene expression in B lymphocytes, leading to their uncontrolled proliferation and transformation [22]. Furthermore, EBV has been reported to cause immune system dysregulation by downregulating the major histocompatibility complex class I and II expression, thus effectively escaping the immune system. Chronic EBV infection can upregulate the checkpoint inhibitors, such as programmed cell death protein 1 (PD-1) inhibitory receptors on the surface of T lymphocytes, leading to poor function of effector T cells, a phenomenon called T-cell exhaustion, which prevents optimal infection control [22]. Upregulation of the ligand of the PD-1 receptor (PD-L1) has also been observed in the setting of EBV infection, further promoting effector T-cell dysfunction and anergy against EBV [22]. Lastly, an association between PTLD and cytomegalovirus (CMV) was recently suggested [16].

It is clear that the biology of EBV-positive versus negative PTLD cases significantly differs from a genetic perspective [12,17]. EBV-positive cases have less frequent and less complex chromosomal molecular abnormalities, whereas EBV-negative cases can be associated with complex karyotypes, typically seen in cases of diffuse large B-cell lymphoma in immunocompetent patients. Genetic studies have also confirmed the resemblance of the latter two entities in a transcriptomic level [12]. 

Despite the clear differences in disease biology, a clear difference in prognosis or response to available therapies between EBV-positive and EBV-negative cases has not been established [25]. However, there have been reports of EBV negativity serving as an adverse prognostic factor. One study demonstrated that the median overall survival (OS) of EBV-negative PTLD patients was significantly lower compared to EBV-positive patients, 1 vs. 37 months, respectively [26,27]. In contrast, a large retrospective analysis from the CIBMRT showed no impact of EBV status on survival [28]. More studies are needed to better define the complex molecular basis of PTLD and create a more accurate classification system with better prognostic ability, as this will allow a more individualized approach to therapy.

## 3. Pathologic Classification 

PTLD is divided into six pathologic subtypes based on the 2017 World Health Organization (WHO) classification (Table 1) [13,29]. Three subtypes are described as nondestructive PTLD and include plasmacytic hyperplasia, infectious mononucleosis-like PTLD, and florid follicular hyperplasia, whereas the three remaining subtypes are mentioned as destructive PTLD and include polymorphic, monomorphic, and classic Hodgkin lymphoma-like PTLD [13]. 

The monomorphic subtype is the most common, accounting for 75% of all PTLD. Since it cannot be distinguished from sporadic B-cell non-Hodgkin’s lymphomas from a pathologic aspect, it follows the sub-classification of non-Hodgkin’s B-cell malignancies. Notably, the most common type of monomorphic PTLD is DLBCL-like PTLD [30,31]. Immunohistochemical and gene-expression analyses have shown that, for DLBCL subtype of PTLD, EBV-negative cases are usually of germinal center origin, whereas EBV-positive cases express a post-germinal center profile [23,32,33,34]. Follicular lymphomas and chronic lymphocytic leukemia (CLL) are not considered PTLD. Mantle cell lymphomas are also not considered PTLD in their vast majority; however, EBV-positive mucosa-associated lymphoid tissue (MALT) lymphoma has recently been included as a type of PTLD in the latest WHO classification [29]. 

Most histologic subtypes have a strong association with EBV, which is observed in almost all cases of nondestructive PTLD. The nondestructive forms are early-onset and less aggressive entities that present with milder symptoms. Poly- or monoclonal subtypes may also be EBV-driven. EBV-unrelated cases typically display monomorphic morphology [15,26,32,35]. The disease subtype appears to have prognostic value. Monomorphic T-cell PTLD is characterized by the lowest overall survival rates in comparison to all other subtypes. The Burkitt subtype of monomorphic B-cell PTLD also appears to be associated with poor OS [36].

## 4. Clinical Features

The incidence of PTLD is bimodal, with early- and late-onset subtypes. Typically, the incidence of PTLD is high within the first year of transplantation (early-onset) in both alloHSCT and SOT recipients [1]. However, given the recent advances in management leading to prolonged survival, late onset PTLD has also emerged after >5 years post-transplant, sometimes as late as >15–20 years post-transplant, a phenomenon most commonly seen in SOT recipients [13,37]. 

Early-onset PLTD is mainly associated with EBV (>90% of cases), whereas late onset PLTD can frequently be EBV-negative [11,12]. A recent study on liver transplantation patients, demonstrated 91% and 66% EBV-positivity in early- and late-onset PTLD cases, respectively [38]. Moreover, 88% of very early PTLD cases after SOT vs. 52% of late-onset PTLD cases post kidney transplantation were found to be EBV-positive [38]. Similar frequencies have been reported in other studies [39,40]. In addition, early PTLD usually presents with nondestructive or polymorphic histology, whereas late onset PLTD is usually monomorphic in nature [41].

Post-alloHSCT PTLD derives from the donor’s B lymphocytes is almost exclusively EBV-related, and occurs during the first year after transplantation, particularly within the first 2–6 months [42]. Late-onset PTLD post-alloHSCT is a very rare entity. In contrast, PTLD after SOT mainly arises from the recipient’s B lymphocytes and can also occur years after transplantation, with these cases typically being unrelated to EBV infection. Usually, any early onset PTLD tends to be EBV-driven and presents with non-destructive or polymorphic pathologic subtypes [1,13].

PTLD is characterized by a heterogeneous clinical presentation, ranging from asymptomatic to life-threatening, including spontaneous tumor lysis and organ failure. Symptoms may include fatigue, malaise, and mononucleosis-like symptoms, or even B-symptoms (fever, weight loss, night sweats, and lymphadenopathy). It often develops rapidly, requiring prompt diagnosis and treatment [1]. Some of the most common locations for disease development are the lymph nodes, tonsils, spleen, and bone marrow but, also, solid organs such as liver, lung, and kidney [7]. In comparison to lymphomas in immunocompetent hosts, it is more often associated with extranodal involvement [13].

Regarding prognosis, EBV status has not been correlated with survival to date. On the other hand, the histologic subtype seems to have an association with nondestructive early-onset forms and polymorphic PTLD having better survival. Early-onset PTLD also appears to have a better prognosis and response to reduction in immune suppressive treatment. Historically, higher mortality rates were reported in post-alloHSCT PTLD [43], compared to post-SOT PTLD [30,35,44]. However, most recently, this paradigm has shifted and outcomes have overall improved with the introduction of rituximab into clinical practice [45,46,47]. Notably, in PTLD where reduction in IST is not feasible, commonly in cases of cardiac transplant, prognosis remains poor [46,48]. 

## 5. Diagnosis and Staging 

The diagnosis of PTLD is established through histopathological examination of tissue specimen obtained with surgical excisional (preferably), incisional or core-needle biopsy [49]. Immunohistochemical staining should be the standard for sporadic lymphomas, including testing for T and B-cell markers to establish the pathologic subtype of the PTLD based on the WHO criteria (Table 1). Fluorescence in situ hybridization (FISH) analysis of lymphoma should be performed to help with further subclassification once a subtype has been confirmed. Although not prospectively validated and not mandatory for diagnosis, an EBV-encoded RNA (EBER) in situ hybridization assay is recommended in all cases of PTLD, due to its association with EBV [50].

After histopathological confirmation, accurate staging is necessary. The same staging guidelines already established for lymphomas in immunocompetent hosts should also be used for PTLD [13]. 18F-fluorodeoxyglucose positron-emission tomography (FDG-PET) in combination with computed tomography (CT) have been suggested as highly sensitive for staging [51]. PET-CT is usually recommended as the initial diagnostic imaging modality. If not available, then CT of the chest, abdomen, and pelvis should be performed instead [49]. Based on disease burden shown in the initial imaging and clinical presentation, further diagnostic studies should be considered, such as magnetic resonance imaging (MRI) or CT of the brain/orbits/sinuses, lumbar puncture in case of neurological symptoms, and/or bone marrow biopsy [10].

## 6. Risk Factors

A significant risk factor for PTLD development, specific to SOT, is the use of multivisceral or intestinal grafts, since they contain an increased load of donor lymphoid tissue, which is at risk for expansion upon EBV infection during an immunocompromised state. On the other hand, kidney grafts have the lowest risk for PTLD [22]. The risk of PTLD is significantly higher among EBV-naïve SOT recipients who acquire primary EBV infection after transplant in the context of IST, which impairs their initial T-cell response and allows unrestricted viral replication. Given that >90% of organ donors have been infected with EBV, most naïve recipients will receive an EBV-infected allograft and will subsequently develop primary EBV infection [52]. Approximately 10% of these patients will eventually develop PTLD, which poses a 10-fold increased risk compared to patients who were seropositive prior to SOT [53,54,55]. Additional risk factors are described in Table 2 [48,56,57,58,59,60,61,62,63,64,65].

Recipients’ EBV immunoglobulin-G (IgG) serology pre-SOT is used to identify high-risk patients. Most EBV serology assays detect antibodies to both viral capsid antigen (VCA) and Epstein–Barr nuclear antigen-1 (EBNA-1), which are concordantly positive in most individuals who have been EBV-infected. However, discordant VCA and EBNA-1 IgG results have been described in a minority of EBV-infected patients. A recent large retrospective study suggested that the risk of PTLD among SOT recipients with discordant VCA and EBNA-1 was approximately threefold that of patients with concordantly positive serologies [66]. Future studies are needed to further investigate those results.

With respect to alloHSCT, the degree of matching between the donor and the recipient, as well as the type of conditioning regiment play a significant role in the development of PTLD. The higher the mismatch (for example, in cases of haploidentical or mismatched unrelated transplants), the higher the need for selective T-cell depletion protocols and IST, hence the higher the risk of PTLD development [67]. The use of reduced-intensity conditioning or anti-thymocyte globulin (ATG) as part of the conditioning regimen are strong contributors to the development of PTLD, the latter in a dose-dependent manner [42,68]. Other risk factors are mentioned in Table 2 [67,69,70,71,72,73].

## 7. Surveillance and Prevention 

A high or rapidly increasing viral load is associated with an increased risk of PTLD [74,75]. Despite the lack of guidelines recommending pre-emptive serial monitoring of EBV viral load early post-transplant, many centers have adopted this strategy in an effort to promptly identify patients at high risk for PTLD requiring early intervention. Monitoring of EBV DNA viral load is usually performed by quantitative polymerase-chain reaction (PCR). 

At present, surveillance is strongly encouraged for the patients with pre- and peri-transplant high-risk factors and/or those who have undergone alloHSCT. The Sixth European Conference on Infections in Leukemia (ECIL-6) consensus recommends starting monitoring EBV levels within the first month after alloHSCT and continue for at least 4 months after, with a weekly frequency at least up until reconstitution of cellular immunity [76]. There is no data to support a preference for whole blood, plasma, or serum; all are acceptable specimens. Despite these recommendations, there are no official guidelines regarding when surveillance should be initiated, who are defined as high-risk patients, the frequency of surveillance, or the threshold to start pre-emptive therapy [22].

Appropriate prevention can help avoid development of PTLD, especially in individuals with high-risk features. In these patients, necessary measures should be carefully implemented to minimize all the potential modifiable risk factors. A major approach is limiting the exposure to aggressive post-transplant IST and tapering of IST post-transplant to the lowest possible level to avoid organ rejection. This is particularly important in patients with increasing levels in circulating EBV DNA. 

In patients undergoing alloHSCT, selection of the best-matched donor and optimal conditioning regimen is important. Incorporation of rituximab in the conditioning regimen or induction immunosuppression for alloHSCT or SOT, respectively, appears to be protective. Avoidance of ATG or specific T-cell depletion protocols can also decrease the risk [77]. When possible, EBV-naïve recipients should avoid EBV-seropositive allografts, as they pose a high-risk situation for development of PTLD [78].

Most recently, vaccine therapy has been considered for patients at risk for PTLD before organ transplantation. A phase I trial (NCT00278200) is currently studying the administration of a photochemically-treated autologous EBV-transformed B lymphoblastoid cell vaccine in EBV-positive or -negative patients who are being considered for SOT and are at high risk for PTLD. The primary objectives are to determine its efficacy in achieving an EBV-specific T-cell and antibody response and whether this is able to prevent primary EBV infection in EBV-negative patients [22].

## 8. Pre-Emptive Therapy

With the widespread incorporation of serial EBV-DNA monitoring, the initiation of pre-emptive therapy at the time of viral reactivation has become a common practice [79,80]. Most institutions reduce IST as a first step, when feasible and safe from an organ rejection perspective; however, there is not an official consensus. In cases where IST reduction is inadequate or not feasible as an initial approach, therapy with rituximab is pursued. There are no guidelines regarding the EBV threshold that should trigger pre-emptive treatment. This is partially due to variability and lack of standardization in source of samples (whole blood, plasma, and serum) and available assays.

Use of rituximab in high-risk patients is highly effective in preventing PTLD in post-alloHSCT patients with rising EBV load [81,82,83,84]. In post-SOT setting, findings are similar; decreased PTLD rates were reported with the use of rituximab post-heart or kidney transplants associated with uncontrolled EBV viremia that did not respond to IST reduction. These findings are attributed to the CD20+ B-cell depletion caused by rituximab, since these cells are the largest reservoir for latent EBV. The reduction in these cells lowers the risk for their malignant transformation and, thus, PTLD [22]. 

Rituximab is currently recommended on a weekly basis for 1–4 doses, and this approach is estimated to decrease the incidence of PLTD in >90% of cases [76]. Less popular approaches include the administration of EBV-specific T lymphocytes from the donor or another human; however, this is a time-consuming process and difficult to apply in daily clinical practice. Other approaches, such as the use of antivirals, have not been proven to be effective [22]. Importantly, all the aforementioned approaches should only be considered for high-risk patients; however, there is no consensus defining “high risk”.

## 9. Treatment

Treatment of PTLD can be heterogeneous, based on the disease subtype, transplant type, and patient characteristics [79]. Notably, common therapeutic strategies for PTLD differ from those used in lymphoproliferative disorders of immunocompetent patients. Most frequently used modalities include IST reduction, rituximab monotherapy or in combination with chemotherapy, and, less frequently, use of other novel immunotherapies, or autologous stem-cell transplantation. Surgery or local radiation can be options for localized disease [13] (Figure 1).

### 9.1. SOT Recipients 

#### 9.1.1. Reduction in IST

IST reduction is usually the first step. Initial management should include dose reduction of at least 50% in calcineurin inhibitors and withdrawal of any antimetabolites (e.g., azathioprine and mycophenolate) [79]. This significantly helps the expansion of the cytotoxic T-cell compartment without completely compromising allograft function. In cases of severe PTLD, all immunosuppressive agents should be temporarily discontinued, except steroids. It is crucial to monitor for allograft rejection while patients receive low-dose IST as acute solid organ rejection has been reported at frequencies reaching 37% [48].

Reduction in IST has variable efficacy, which can be as high as 80%. However, cases of EBV-negative PTLD, high burden or disease, or advanced stage are all factors leading to poor response to IST reduction, emphasizing the need for more aggressive approaches in these patient populations [48].

#### 9.1.2. Rituximab

In cases of no or inadequate response to IST reduction, or when IST reduction is not feasible due to a high risk for allograft rejection, the chimeric murine/human IgG1 kappa anti-CD20 mAb, rituximab, is used as frontline therapy [1]. For rare histologic subtypes, treatment strategy is different and will be discussed later. Rituximab binds the CD20 antigen found on the surface of B lymphocytes and eradicates them via complement- or antibody-dependent cell-mediated cytotoxicity. Rituximab can be administered either as a monotherapy or in combination with chemotherapy and has shown adequate clinical efficacy up to 60%, based on type, stage, and bulk of disease. The most used chemotherapy regimen is cyclophosphamide, doxorubicin, vincristine, and prednisone (CHOP), which is the typical frontline regimen given for B-cell lymphomas in immunocompetent hosts.

Two phase II trials assessed the administration of four weekly doses of frontline rituximab in post-SOT patients who failed IST reduction. The first trial included 17 patients, with 53% achieving complete response (CR). All responders had EBV-related PTLD. The 3-year OS rate was 56%. All patients who did not benefit from rituximab had EBV-negative and late onset PTLD, and almost all required subsequent chemotherapy, suggesting that EBV-negative PTLD might need combination of rituximab and chemotherapy from the beginning [85]. The second trial evaluated 43 patients, and showed an overall response rate (ORR) of 44% to rituximab monotherapy, with 28% achieving CR. The 1-year PFS and OS rates were 21% and 67%, respectively. [86]. Another phase II trial, assessed the efficacy of upfront rituximab with concomitant IST reduction in 38 patients as a first-line approach. Those who did not achieve CR received a second course of four rituximab infusions. CR and PR rates were 34% and 45%, respectively, after the first course of rituximab. Retreatment of patients achieving PR with rituximab yielded a CR rate of 83%. Those who had no response (38%) to the first course were treated with chemotherapy, and 75% of them achieved CR. At 27.5 months, PFS and OS rates were 42% and 47%, respectively [87]. A long-term follow up of this trial, as well as a real-world cohort of 21 patients treated with the same regimen after the trial was closed (validation cohort), was recently published. For the trial patients (median follow-up of 13 years), the disease-specific survival (DSS) at 10 years was 64.7%; for those that achieved CR (61%), DSS at 5 and 10 years was 94.4% and 88.1%, respectively. For the real-word cohort (median follow-up of 6.5 years), DSS at 5 years was 75.2%; for those that achieved CR (38%), DSS was 87.5%. Authors concluded that PTLD patients in CR after rituximab have an excellent long-term outcome, reproducible in the real-world setting [88].

#### 9.1.3. Rituximab and Chemotherapy

Transplant patients have historically tolerated conventional chemotherapy poorly, with a treatment-related mortality (TRM) of up to 31%, compared to 2% for DLBCL in immunocompetent individuals [86,89,90]. Despite this, patients capable of tolerating chemotherapy may achieve a long-lasting remission as outlined from the PTLD-1 and PTLD-2 trials. 

The PTLD-1 trial (NCT01458548) assessed the upfront use rituximab monotherapy followed by CHOP consolidation in 70 patients who were refractory to initial IST reduction. ORR and CR were 60% and 20%, respectively, after rituximab induction, as well as 90% and 68%, respectively, after CHOP consolidation. The 3- and 5-year OS rates were 65% and 57%, respectively, with median PFS and OS of 4 and 6.6 years. Grade 3–4 toxicities were high and 11% had CHOP-TRM [91,92]. Response to rituximab induction was a prognostic factor for improved OS. 

Given the high efficacy of rituximab but increased CHOP-TRM, a change in protocol was made (PTLD-1/3, 3rd amendment; NCT00590447) and a response-adapted treatment strategy was introduced to minimize toxicity: rituximab consolidation for complete responders to rituximab induction, and R-CHOP consolidation for all others. Among 152 participants, 25% achieved CR to single-agent rituximab. For those who achieved less than CR, ORR with subsequent R-CHOP was 85%, with a 60% CR rate. ORR and CR for the entire cohort were 88% and 70%, respectively. Median time to progression (TTP) was not reached, and the 3-year estimate without progression was 78%. Median OS was 6.6 years. Response to rituximab induction was a highly significant predictor of better OS, TTP, and PFS (*p* < 0.001). TRM was lower at 7% [93,94]. 

A follow-up phase II trial, the PTLD-2 (NCT02042391) (*n* = 48), applied a slightly modified risk-stratification treatment approach, based again on response to rituximab induction and international prognostic score (IPI) score at diagnosis. Low-risk patients, defined as those achieving CR or RP with IPI < 3, received a repeat course of rituximab as consolidation. The rest received R-CHOP or more intensifying chemotherapy regimens (Table 3). ORR to rituximab induction was 45% but only 9% achieved CR. The ORR at final staging was 94% and median PFS 3.8 years, with 2-year PFS and OS rate of 56% and 68%, respectively. Given that comparable outcomes to the PTLD-1 were yielded with a less intense regimen for complete responders, a stepwise approach should be followed to avoid unnecessary toxicity in these patients [95]. Rituximab has also been combined with more intense chemotherapy regimens, such as etoposide, prednisone, vincristine, cyclophosphamide, and doxorubicin (EPOCH) in small case series; however, clear survival benefit has not been observed [22,37,96].

### 9.2. AlloHSCT Recipients 

Data for frontline therapy in patients post-alloHSCT is very limited [100]. Reduction in IST is challenging due to the high risk of graft versus host disease and appears to have very limited efficacy in early-onset PLTD. This is because of different mechanisms of the underlying cytotoxic T-cell compartment weakening, compared to post-SOT PTLD. In particular, the donor’s immune system needs time to fully recover and expand within the recipient’s marrow, with early-onset PTLD being a consequence of this T-cell insufficiency.

Frontline rituximab is typically used for early-onset PTLD displaying the most common histologic subtypes, with efficacy around 60%. Upfront chemotherapy, most frequently CHOP, can also be used along with rituximab if high burden of disease is noted. Alternatively, if inadequate response to rituximab monotherapy is noted, consolidation R-CHOP can be used. Prospective or observational large population studies are not available for this rare entity; therefore, decision making is largely institution-dependent [101]. 

A retrospective study explored the prognostic factors that affect the outcome of EBV-related PTLD after rituximab-based therapy in post-alloHSCT patients. Poor response to rituximab was noted in patients with age ≥ 30 years, extra-lymphoid tissue involvement, acute GVHD, and a lack of reduction in IST upon PTLD diagnosis. Notably, IST tapering was associated with a lower PTLD mortality (16% vs. 39%) and a decrease in EBV DNA levels during therapy was predictive of better survival [102]. A recent systematic review reported that the percentage of patients who fail rituximab post-alloHSCT varies greatly (13–67%) but outcomes are overall poor [103]. 

### 9.3. Treatment of Relapsed Disease

There are no established treatments for relapsed PTLD where all first-line options (IST reduction, rituximab, and CHOP) have been utilized. In this context, decision regarding treatment is institution-dependent based on experience and tumor characteristics. Sometimes based on the histologic type, regimens used for relapsed lymphoma in immunocompetent patients may be considered. These include but are not limited to ifosfamide, carboplatin, and etoposide, or gemcitabine and oxaliplatin; however, these regimens have increased toxicity which PTLD patients have difficulty handling. Enrollment in clinical trials is always encouraged under these circumstances. 

A pilot study assessed the efficacy and safety of carboplatin/etoposide in nine patients with relapsed-refractory PTLD post-SOT who were not candidates for intensified salvage regimens. Five patients achieved a CR and one SD, suggesting that carboplatin/etoposide might be a combination with some degree of effect [104]. In another case series of seven patients, some of whom had received salvage with rituximab several times, retreatment with salvage rituximab achieved CR in three patients and PR in one. Median PFS was nine months [105]. These data suggest that rituximab salvage therapy could be effective for intensively pretreated patients [49]. 

Another available modality is autologous hematopoietic cell transplantation (autoHSCT); however, PTLD patients have an increased risk for complications and death, making autoHSCT transplant unfeasible in most cases. At present, published data on the efficacy and safety of auto-HSCT in PTLD are limited to case reports and series. A retrospective analysis of 21 patients who received autoHSCT for relapsed PTLD (mainly DLBCL-type, median of two prior lines of therapy) following SOT demonstrated CR and PR of 47% and 38%, respectively. The 3-year PFS and 3-year OS were 62% and 61%, respectively. Overall, 12 deaths were reported, including 4 related to autoHSCT. The 100-day non-relapse-mortality (NRM) and 1-year NRM were 14% and 24%, respectively. The authors concluded that autoHSCT is a feasible option for relapsed PTLD; however, due to the high associated infectious-driven NRM, careful selection is critical [106].

### 9.4. Novel Approaches 

#### 9.4.1. EBV-Specific Cytotoxic Lymphocytes 

In cases of EBV-positive PTLD, EBV-specific cytotoxic lymphocytes (CTLs) appear capable of inducing a robust EBV-targeted T-cellular immune response. EBV-CTLs can be used for prevention, pre-emptive therapy, and front-line treatment, and they are usually donor or HLA-matched third-party-derived. They can be combined with rituximab, either as a first step or in a stepwise fashion with or without rituximab in patients who have yielded low or no responses to upfront rituximab. 

Tabelecleucel, a third-party EBV-CTL product derived from volunteer donors (partially HLA-matched, sharing ≥2 HLA alleles), was evaluated in patients with EBV-positive PTLD. In a phase I/II study (NCT00002663 and NCT01498484) of 46 patients with EBV-positive PTLD refractory to frontline rituximab, tabelecleucel yielded an ORR of 68% (post-HSCT) and 54% (post-SOT), with no significant toxicities. For patients who achieved CR/PR (responders), 1-year OS rate was high at 89% and 82%, respectively [97]. At present, there is one ongoing phase III trial (ALLELE, NCT03394365) assessing the use of tabelecleucel in patients with EBV-positive PTLD who have failed front-line rituximab or both rituximab and chemotherapy. An interim analysis showed an ORR of 50% on both post-SOT and post-HSCT patients, with median overall OS of 18.4 months and 1-year OS rate of 61%. Notably, responders appear to have better survival outcomes compared to non-responders (Table 3) [98,99].

An early phase I study is assessing the efficacy of tacrolimus-resistant EBV-CTLs for PTLD post-SOT (ITREC, NCT03131934). Several other early trials are currently exploring the role of EBV-CTLs (NCT01555892, NCT02580539, and NCT02822495) or their combination with other agents, such as brentuximab vedotin for rituximab-refractory disease.

#### 9.4.2. Antibody Drug Conjugates

Ibritumomab tiuxetan, an anti-CD20 mAb conjugated with a radioactive isotope, was assessed in a series of eight patients post-SOT with relapsed-refractory PTLD (median of two prior lines of therapy), as a single agent (*n* = 7) or with chemotherapy (*n* = 1) and yielded an OS rate of 62.5% (all CR) after a median follow-up of 37 months [107].

The anti-CD30 antibody drug conjugate brentuximab vedotin (BV) is currently approved in combination with doxorubicin-based chemotherapy (BV-CHP) as a front-line therapy for CD30+ peripheral T-cell lymphomas (PTCL), after the completion of the phase III pivotal ECHELON-2 trial, where BV-CHP was found to be superior to CHOP. BV is now being investigated in the setting of PTLD, given that there is increasing evidence showing that the CD30+ antigen expression is frequently detected in PTLD [108,109,110]. ECHELON-2 excluded immunocompromised patients, given their poor survival outcomes. However, recently there has been increasing interest in the use of BV-CHP in CD30+ T-cell PTLD patients due to poor responses to rituximab and/or CHOP. One case reported a very prolonged response to BV-CHP in a patient with CD30-positive T-cell PTLD [4], while BV together with EBV-specific cytotoxic T lymphocytes (EBV-CTLs) resulted in CR lasting for >3.5 years in a patient with progressive EBV-positive PTLD post-alloHSCT [111]. 

A phase I-II trial (NCT01805037) investigated the combination of BV with rituximab as front-line therapy in 22 patients with immunosuppression-associated CD30+ and/or EBV-positive lymphoid malignancies, 16 of which were diagnosed with PTLD. The combination resulted in a CR rate of 60% after a median follow-up of 26 months, with 1-year and 3-year PFS rates of 75% and 67%, respectively; however, the study was terminated due to lack of funding [112]. Another phase II study (NCT04138875) assessing a risk-stratified sequential treatment approach of PTLD, with BV and rituximab plus or minus bendamustine (triplet therapy for high-risk patients), was also withdrawn due to the lack of funding. 

#### 9.4.3. Targeted Therapies

The Bruton tyrosine kinase (BTK) inhibitor, ibrutinib, has been studied with rituximab and CHOP prospectively in the phase II TIDaL trial that uses a risk-stratified sequential approach. The combination of ibrutinib and rituximab with or without chemotherapy (based on a risk-stratified strategy) was evaluated in 39 patients with PTLD post-SOT. All patients received ibrutinib daily with four doses of rituximab. Interim response was assessed after ~6 weeks and patients achieving CR/PR continued with the same regimen for four further 3-weekly doses of rituximab with concomitant ibrutinib. Patients who did not respond received four cycles of R-CHOP and ibrutinib instead. After the initial rituximab–ibrutinib combination, 29% achieved CR which was not high enough to warrant further investigation [113]. Acalabrutinib, another BTK inhibitor, is currently studied in combination with rituximab for newly diagnosed PTLD in a phase II trial (NCT04337827).

#### 9.4.4. Antiviral Therapies

B cells latently infected with EBV and EBV(+) lymphoproliferative disorders do not express the viral thymidine kinase and thus are unaffected by antiviral agents, such as purine nucleoside analogs. Therefore, monotherapy with nucleoside analogs does not induce any positive responses in EBV-positive PTLD. Pharmacological attempts of inducing the expression of the viral thymidine kinase through the administration of the histone deacetylase inhibitor, arginine butyrate, before administering antivirals, have led to promising results with acceptable toxicity [114]. Similarly, immunomodulatory drugs or proteasome inhibitors appear capable of inducing EBV lytic activation, enhancing the efficacy of antiviral agents [115,116]. There is an ongoing phase Ib/II study assessing the safety and efficacy of the histone deacetylase inhibitor VRx-3996 in combination with valganciclovir in patients with EBV-related lymphomas (NCT03397706). At the same time, novel antivirals are currently being developed. One example is the new antiviral agent, hexadecyloxypropyl-cidofovir (HDP-CDV) that seems to exhibit a significant increase in antiviral activity against various viruses, including EBV in vitro [117]. Further results are eagerly awaited.

#### 9.4.5. Serotype-Dependent Recombinant Adeno-Associated Vector (AAV)

Recombinant adeno-associated virus (rAAV) utilization has raised the prospect of novel, focused, and effective therapy. Although predominantly associated with gene replacement, advances in altering the tropism of viruses, as well as the content and structure of the viral genome have led to an increasing interest in using rAAV for precision cancer therapies [118]. Initial studies erroneously suggested that human B cells are not susceptible to infection with rAAV; however, newer rAAV variants can now be used for the treatment of B-cell tumors and help eradicate focal forms of PTLD [119]. In detail, B-cell infection with EBV increases transduction susceptibility, with rAAV6.2 and its closely related serotypes, rAAV6 and rAAV6TM, displaying the maximal transduction efficiency among 15 examined serotypes. Furthermore, preincubation of rAAV cells with complexed CD40 ligand (CD40L, a member of tumor necrosis factor family involved in B-cell activation and antibody production) or anti-IgM antibody appears to augment rAAV2 transduction capabilities through an unclear mechanism [120].

In this context, apart from the therapeutic potential of rAAV6.2, it has been suggested that it may also be used to introduce nucleic acids to tumor B cells that are difficult to transfect, aiming to uncover oncogenic mechanisms. Unfortunately, rAAV6.2 appears to be capable of transfecting other types of human cells in addition to B cells, indicating that a more specific mutational rAAV6.2 capsid analysis is necessary in order to identify the amino acid substitutions that will guarantee a more selective tropism and, thus, a lower side effect profile at clinical application [119].

#### 9.4.6. T-Cell-Redirecting Bispecific Antibodies

Blinatumomab is a T-cell-redirecting bispecific antibody or bispecific T-cell antigen engager antibody (BiTE) that forms an immunologic bridge between the T cells (via binding to the CD3 receptor) and tumor cells (via binding to the CD19+ antigen), leading to the destruction of the latter. So far, it has been used successfully for early relapsed acute B-cell lymphoblastic leukemia. A recent case report described its use in the context of relapsed EBV-related PTLD post-alloHSCT in a pediatric patient whose tumor displayed DLBCL histology and expressed CD19. After blinatumomab administration, no cytokine release syndrome (CRS) or other side effects were noted and the patient demonstrated rapid improvement with declining EBV titers, with no evidence of relapse after a follow-up of 7 months, suggesting further assessment of blinatumomab in adult PTLD [121].

#### 9.4.7. Chimeric Antigen Receptor (CAR) T-Cell Therapies

The use of CD19 chimeric antigen receptor T-cell (CAR-T) therapy is currently being investigated in the setting of PTLD, after having shown remarkable efficacy in relapsed-refractory DLBCL in immunocompetent patients. However, its application in PTLD has some theoretical barriers. These mainly include the continuous IST that these patients receive, which may compromise the ability for T-cell collection at leukapheresis and the CAR T-cell expansion at the time of infusion [122].

Despite these concerns, CAR T-cell therapy was recently used in a kidney transplant patient with PTLD refractory to immunochemotherapy. IST was discontinued before the infusion to best allow CAR T-cell expansion, and the patient was closely monitored for early signs of organ rejection. CAR T-cell therapy led to CR, and the patient remained off IST thereafter, even 7 months post-infusion. In another report, a kidney transplant patient with high-burden refractory PTLD was enrolled in a clinical trial of CAR T-cell treatment (*ChiCTR1800019622*) combined with PD-1 inhibitor induction, followed by PD-1 inhibitor maintenance therapy. Here, IST was not discontinued and the patient achieved PR [123].

Recently, further case series have demonstrated the safety and feasibility of CAR T-cell therapy. A case series from MD Anderson reported that three patients were treated with the anti-CD19 CAR T-cell therapy, axicabtagene ciloleucel (axi-cel), for stage IV DLBCL-like PTLD, all 7–10 years post-kidney transplant. All patients were refractory to front-line and/or salvage immunochemotherapy. IST was discontinued 2–4 weeks prior to leukapheresis and remained off post-CAR T-cell infusion. Two patients achieved CR, whereas one patient PR. Two patients eventually relapsed (at week +34 and +12, respectively). Toxicity was variable with CRS only at grade 1; one patient developed neurotoxicity grade 3. One patient had organ rejection, given ongoing discontinuation of IST post-CAR T-cell therapy. Authors concluded that axi-cel is feasible in kidney transplant recipients with DLBCL and suggested that IST can be safely stopped 2 to 4 weeks before leukapheresis and may be reinitiated 4–12 weeks after axi-cel in patients with ongoing remission, with close monitoring of kidney function [124].

Another recent case series described poor outcomes with axi-cel in patients with DLBCL-like PTLD, 10–20 years after SOT (heart, kidney, and pancreas, respectively), which was refractory to first- or second-line chemotherapy. All patients developed complications, such as CRS and neurotoxicity requiring tocilizumab, and 2/3 patients developed severe acute renal injury (AKI) requiring renal replacement therapy. The patient who developed AKI expired after withdrawal of care due to lack of response and toxicity. [125].

Tisagenlecleucel (tisa-cel), another anti-CD19 CAR T-cell product, was used for the management of three patients with refractory EBV-negative DLBCL-like PTLD. All patients continued calcineurin inhibitors throughout the whole course of treatment and responded to a single infusion of tisa-cel, with two achieving CR. Toxicity profile was similar to other patients with non-PTLD DLBCL treated with tisa-cel [126].

Despite the encouraging outcomes, it appears that CAR T-cell therapy can result in high rates of toxicity, and to date there are no reliable predictors of either toxicity occurrence or disease response. Additionally, the impact and timing of restarting IST post-CAR T-cell therapy has yet to be determined. Further prospective studies are warranted to answer these important questions, which will help patient selection with identification of a subgroup of PTLD patients that would receive the greatest benefit and experience the least possible risk for CAR T-related toxicity [127].

### 9.5. Radiation Therapy

Radiation therapy, either alone or in combination with other treatments, is another modality that can be used in PTLD patients who present with localized disease or CNS involvement. Localized forms of PTLD could be efficiently not only treated, but also possibly cured with radiotherapy, given the excellent radio-sensitivity of lymphoid neoplasms. In one report, three adult patients with PTLD were successfully treated with moderate-dose (24–36 Gy) radiotherapy, achieving sufficient local control with minimal toxicity [128]. In another example, ultra-low-dose radiation was used for ocular MALT-subtype PTLD and led to a sustained treatment response [127].

### 9.6. Treatment of Rare Histologic Subtypes

PTLD can rarely display uncommon histologies, such as primary central nervous system (CNS) lymphoma, Burkitt lymphoma, plasmacytic or plasmablastic types, and classic Hodgkin lymphoma, that require different approaches. There is scarce evidence-based information on how to treat these subtypes; hence, guidelines for lymphomas in immunocompetent patients are usually followed. The benefit of reduction in IST is unclear, however, it should be strongly considered as a first step in combination with immunochemotherapy.

#### 9.6.1. CNS PTLD

Evens et al. performed a large retrospective analysis (*n* = 84) of patients with CNS lymphoma, mainly with DLBCL-like histology [129]. First-line regimens included high-dose methotrexate (48%), high-dose cytarabine (33%), rituximab (45%), and whole-brain radiation, either alone or in various combinations, with an ORR of 60%. Treatment-related toxicity and mortality were high, with the latter reaching 13%. Most patients (93%) had reduction in IST as part of their management. After a median follow-up of 42 months, 3-year PFS and OS rates were 32% and 43%, respectively. There was a trend for improved PFS for patients who received rituximab and/or high-dose cytarabine. Poor performance status predicted inferior PFS, while increased LDH portended inferior OS. A retrospective study of 25 pediatric patients (84% post-SOT) demonstrated favorable outcomes following systemic and intrathecal chemotherapy and rituximab (4-year PFS and OS were 50% and 74%, respectively) [130]. Another study of 14 adult patients with CNS-PTLD demonstrated that intrathecal rituximab was effective for CNS-PTLD in post-alloHSCT patients who did not respond to frontline intravenous rituximab-based regimens [131]. Recent case reports have described that ibrutinib, with or without third-party EBV-specific CTLs can lead to durable remissions [132].

#### 9.6.2. Burkitt-like PTLD

Burkitt-like PTLD (BL-PTLD) is another rare and aggressive form of PTLD. A retrospective analysis of 23 patients (21 post-SOT) reported that most cases occurred late, after a median of 5.7 years post-transplantation, and were disseminated (stage IV) at diagnosis. Most patients received immunochemotherapy, with 70% achieving CR. All patients treated with rituximab alone required further chemotherapy to achieve CR. The 2-year OS rate was 65%; 75% for patients receiving rituximab-based therapy, and 43% for those without rituximab. Authors concluded that immunochemotherapy combinations (such as R-CHOP or more intensified regimens) yielded good responses, whereas rituximab alone did not, suggesting the use of immunochemotherapy as a first line [133]. Other smaller case series have also described characteristics and outcomes of patients with BL-PTLD, where most patients harboring *MYC* rearrangement had late onset disease associated with EBV. The first included five patients who were managed with reduction in IST, followed by rituximab and/or intensive chemotherapy [134]. The second also included five patients treated under the PETHEMA ALL-3/97 protocol. Despite the encouraging ORR, toxicity was high, especially to those treated with PETHEMA ALL-3/97 regimen [135]. In the last case series, most patients received sequential immunochemotherapy (rituximab followed by CHOP), resulting in CR of 100%, without treatment-related deaths [136]. One patient with CNS disease received additional intrathecal chemotherapy. Given the rarity of BL-PTLD and the data scarcity with regard to therapy outcomes, there are no established treatment guidelines at present. A major challenge as described by Dierickx et al. is the need for prophylactic intrathecal chemotherapy, especially in cases with aggressive and bulky disease [137].

#### 9.6.3. Plasmacytic PTLD

Plasmacytic PTLD is an uncommon variant that accounts for ~4% of all PTLD cases post-SOT. Histologically, it resembles multiple myeloma and frequently presents with extramedullary manifestations. Small case series have reported that reduction in IST and radiotherapy have yielded significant responses in low-burden disease; however, in cases of advanced disease, traditional plasma-cell-directed therapies should be used [138,139]. A small case series evaluated the novel anti-CD38 monoclonal antibody daratumumab [140] in combination with traditional antimyeloma therapy in five post-SOT recipients (4/5 with light-chain amyloidosis), yielding significant responses without organ damage and without always necessitating reduction in IST [141].

#### 9.6.4. Plasmablastic PTLD

Post-transplant plasmablastic lymphoma (PL-PTLD) is an extremely rare type of PTLD. A study of 11 patients revealed morphological and immunophenotypic heterogeneity, with 55% being EBV-positive and 55% harboring MYC rearrangement. Recurrent mutations in epigenetic modifiers, such as the *KMT2/MLL* family of histone H3 methyltransferases were among the most frequent alterations. Treatment was variable, mainly including rituximab and/or chemotherapy (sometimes with addition of bortezomib) with or without radiation; however, given the heterogeneity, no conclusions can be made [142]. Another case series of eight patients combined IST reduction with systemic chemotherapy; however, five patients died from early progression. Three patients achieved and maintained CR, all of whom were EBV-positive, had no cytogenetic aberrations, and received CHOP-21 along with IST reduction [143].

#### 9.6.5. Classic Hodgkin Lymphoma (HL)-like PTLD

Little is known about this uncommon subtype of PTLD. A comparative analysis of 192 patients with HL-PTLD from the Scientific Registry of Transplant Recipients and 13,847 patients with Hodgkin lymphoma in Surveillance, Epidemiology, and End Result (SEER) revealed that HL-PTLD patients had inferior OS. Treatment with HL-specific chemotherapy was associated with improved OS compared to nontraditional HL regimens. In multivariable analysis, advanced age and elevated creatinine were associated with inferior OS [144]. A case series of 13 patients showed superior responses with Adriamycin, Bleomycin, Vinblastine, and Dacarbazine (ABVD)-like chemotherapy than with rituximab; however, toxicity was significant [145]. Smaller case series and case reports also reported poor outcomes with rituximab or non-HL chemotherapy [146,147].

## 10. Conclusions

In summary, PTLD is a rare and severe complication of recipients of alloHSCT or SOT, mainly occurring because of therapeutic IST, and with a large number of cases being strongly associated with EBV infection. Despite recent advances in therapy and prevention, there is still a large gap in evidence-based approaches to treatment given the lack of large prospective clinical trials. Novel therapeutic modalities, increasingly used in immunocompetent hosts with lymphoma, may play a role in the therapeutic landscape of PTLD; however, careful investigation is warranted first given that PLTD has substantial differences compared to lymphoma in immunocompetent hosts, including increased vulnerability and toxicity risk. New insights into the pathophysiology of PTLD are key for the development of a more clinically applicable classification system, as well as development of subtype-specific strategies to improve patient outcomes. Given the rarity of PTLD, multi-institution collaborations are critical to allow the development of phase II and III clinical trials to evaluate treatment interventions in a large number of cases. This will help establish disease-specific guidelines, both preventive and therapeutic, an urgent unmet need in PTLD.

## Figures and Tables

**Figure 1 cancers-14-05949-f001:**
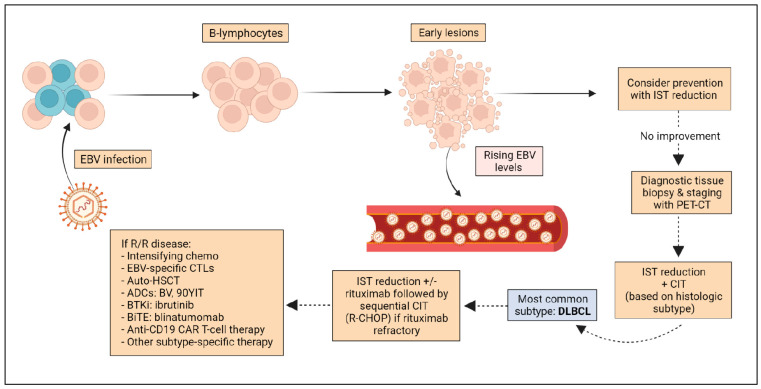
Pathogenesis and treatment algorithm of PLTD. Abbreviations: IST, immunosuppressive therapy; CIT, chemoimmunotherapy; DLBCL, diffuse large B-cell lymphoma; R/R, relapsed/refractory, CTLs, cytotoxic T-lymphocytes; HSCT, hematopoietic stem cell transplant; ADCs, Antibody Drug Conjugates; BV, Brentuximab Vedotin; 90YIT, 90 Yttrium ibritumomab tiuxetan; BTKi, Bruton Tyrosine Kinase Inhibitors; BiTE, Bispecific T cell Engagers; CAR, Chimeric Antigen Receptor.

**Table 1 cancers-14-05949-t001:** Pathologic World Health Organization classification of PTLD.

Pathologic Subtype of PTLD	Location	LN Architecture	Latency from Transplant	Clonality	Further Histologic Subclassification
**1. Non-destructive forms** i. Plasmacytic hyperplasia ii. Infectious Mononucleosis-like PTLD iii. Florid follicular hyperplasia	Nodal	Preserved	Almost all cases early onset	Polyclonal	
**2. Polymorphic PTLD**	Nodal and Extranodal	Destructed	Typically early onset	Polyclonal	
**3. Monomorphic PTLD**	Nodal and Extranodal	Destructed	Both early and late onset	Monoclonal	Per WHO criteria for NHL (most common type is DLBCL)
**4. Classic Hodgkin lymphoma like-PTLD**	Nodal and Extranodal	Destructed	Both early and late onset	Monoclonal	Per WHO criteria for HL

Abbreviations: LN, lymph node; NHL, non-Hodgkin’s lymphoma; DLBCL, diffuse large B-cell lymphoma; HL, Hodgkin’s lymphoma.

**Table 2 cancers-14-05949-t002:** Risk factors for PTLD.

Post-SOT	Post-alloHSCT
**Strong Evidence:**	**Strong Evidence:**
1. Type of Graft:	1. High degree of HLA mismatch
Intestinal > Lung > Heart > others	HLA-mismatched or unrelated donor
Multivisceral grafts or graft from deceased donors	Haploidentical donor
2. EBV Seronegative/naive EBV recipient pre-SOT	Umbilical cord blood graft use
3. High intensity IST	2. Type of conditioning regimen
4. Anti-thymocyte globulin use as part of induction IST	T-cell-depleting strategies (in vivo and ex vivo)
**Weak Evidence:**	Anti-thymocyte globulin use
a. Non-white ethnicity	Non-myeloablative conditioning regimens
b. Young recipient and old donor age	3. Recipient old age > 50 years
c. Non-EBV infection	**Weak Evidence:**
d. Recipient HLA-A26 and B38 status	a. Acute GVHD
	b. History of splenectomy
	c. Diagnosis of Aplastic Anemia
	d. Non-EBV infection

**Table 3 cancers-14-05949-t003:** Selected completed or ongoing clinical trials exclusively for PTLD patients.

NCT Number (Name)	Phase	N	Study Population & Type of Disease	Therapeutic Regimen	Outcomes
NCT01458548 (PTLD-1 trial) [91]	II	70	Newly diagnosed CD20+ PTLD	R induction, followed by CHOP consolidation	▪ *R induction:* ORR 60%, CR 20% ▪ *CHOP consolidation:* ORR 90%, CR 68% ▪ 3 and 5-yr OS: 65% and 57% ▪ mPFS 4 mo, mOS 6.6 years ▪ CHOP-TRM 11%
NCT00590447 (PTLD-1/3) [93,94]	II	152	Newly diagnosed CD20+ PTLD	R-induction followed by response assessment to determine consolidation: **(response adapted strategy)** ▪ CR to R-induction: repeat R ▪ less than CR to R-induction: R-CHOP	▪ *R-induction:* CR 25% ▪ R-CHOP *consolidation*: ORR 85%, CR 60% ▪ Entire cohort: final ORR 88%, CR 70% ▪ 3-yr TTP 78%, mOS 6.6 years ▪ TRM 7%
NCT02042391 (PTLD-2 trial) [95]	II	48	Newly diagnosed CD20+ PTLD	R-induction followed by response assessment to determine consolidation: **(response adapted strategy)** ▪ Low risk (CR or PR w/ IPI 0–2) group: repeat R, *n* = 21 ▪ High risk (PR w/ IPI 3–5 or SD or PD but not heart or lung recipient) group: R-CHOP, *n* = 22 ▪ Very high risk (heart, lung or multiorgan recipients with PD) group: alternating R-CHOP and R-DHAOx, *n* = 5	▪ *R-induction:* ORR 45%, CR 9% ▪ *Entire cohort outcomes at final staging:* ORR 94%, 2-yr OS 68%*,* mPFS 3.8 yrs, 2-yr PFS 56%*,* 2-yr TTP 78%, TRM 7% ▪ *Outcomes by group at final staging:* ***- Low risk:*** ORR 95%, 2-yr EFS 66%, 2 & 3-yr both TTP & PFS 85%, 2 & 3-yr OS 100%, no TRM ***- High risk:*** ORR 100%, 2- and 3-yr TTP 81%, 2 & 3-yr PFS 54%, 2 & 3-yr OS 59%, Grade 3/4 toxicity 50%, TRM 8% ***- Very high risk:*** ORR 60%, CR 40%, 2-yr TTP 33%, 2-yr PFS 11%, mOS 7.4 m, 2-yr OS 30%, grade 3/4 toxicity 63%, TRM 25%
NCT01498484 NCT00002663 [97]	II	46	EBV (+) PTLD patients who have failed frontline R -*n* = 33 post alloHSCT -*n* = 12 post SOT	EBV-specific T cells CTLs from normal HLA-compatible or partially-matched third-party donor	▪ >PR: 68% (alloHSCT), 54% (SOT) ▪ 1-yr OS for pts who responded (CR/PR) to cycle 1: 88.9% ▪ 1-yr OS for pts with SD post cycle 1: 81.8% ▪ PD: 5 patients, 3/5 responded to CTLs from other donors and achieved CR or durable PR, 1-yr OS 100% ▪ Maximum response: after a median of 2 cycles ▪ No significant toxicity
NCT03394365 (ALLELE) [98,99]	III	66	EBV(+) PTLD patients who have failed frontline R and chemotherapy -*n* = 14 post alloHSCT -*n* = 24 post SOT	Tabelecleucel after R or R + chemotherapy failure in post-SOT and post-alloHSCT patients	Interim results of 38 patients (May 2021): ▪ ORR: 50% (50% post-SOT and 50% post-HSCT) ▪ CR: *n* = 5 post-SOT, *n* = 5 post-HSCT ▪ Overall mTTR 1.1 mo ▪ 11/19 responders had DOR >6 mo, and mDOR was NR ▪ mOS: 18.4 mo (entire cohort), 16.4 mo for post-SOT, and NR for post-HSCT patients. ▪ 1-yr OS: 61.1% (entire cohort), 57.4% for post-SOT, and 66.8% post HSCT patients. ▪ Responders had longer OS vs. non-responders: mOS: NR vs 5,7 mo, 1-yr OS 89.2% vs 32.4% ▪ Serious TEAEs: 62.5% post-SOT and 57.1% post-HSCT; Fatal TEAEs: 16.7% post-SOT and 7.1% post-HSCT, none from study treatment.
NCT04337827	II	62	Newly Diagnosed B-cell PTLD	Rituximab and Acalabrutinib	Ongoing, no available results
NCT02900976	II	18	Newly Diagnosed B-cell PTLD	Rituximab and Allogeneic LMP1/LMP2-Specific Cytotoxic T-Lymphocytes	Ongoing, no available results
NCT03131934	I	18	EBV(+) PTLD post-SOT	Tacrolimus-Resistant EBV CTLs	Ongoing, no available results
NCT04989491 (REPLY)	IV	120	EBV(-) recipients who receive EBV(+) kidney allograft	Single dose of R (375 mg/m2) for prevention of PTLD	Ongoing, no available results

Abbreviations: R, rituximab; ORR, overall response rate; CR, complete response; PR, partial response; SD, stable disease; PD, progressive disease; mPFS, median PFS; mOS, median overall survival; mo, months; TTP, time to progression; EFS, event-free survival; yr, year; IPI, international prognostic index; TRM, treatment-related mortality; mTTR, median time-to-response; mDOR, median duration-of-response; NR, not reached; TEAEs, treatment; alloHSCT, allogeneic stem cell transplant; LMP, latent membrane proteins.

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
