# Peer review of "Recent Advances in Adult Post-Transplant Lymphoproliferative Disorder"

_cancers, 2022, doi:10.3390/cancers14235949_

Round 1

Reviewer 1 Report

The authors reviewed PTLD from pathogenesis to recent advance studies. 

1. In table 1, the text in "Further diagnosic information" should be modified for clear message.

Author Response

Please see comments addressed in the pdf file attached. 

Reviewer 2 Report

The authors submitted a nice and comprehensive overview on PTLD.

I have some minor remarks/suggestions:

* Abstract:

- line 33-34: stepwise chromosomal and genetic alterations is mainly seen in EBV-negative causes, whereas EBV-positive cases are less complex. This should be corrected (Finalet Ferreiro J, et al. Am J Transplant 2016;16:414-25 = ref 31).

- line 40: "therapy is similar to..." : this is not true and contrasts with the message of the authors later in the manuscript.

* Introduction:

- line 69-70: "so far, a combination...": this is not completely true as sometimes rituximab monotherapy can be used. It is true if chemotherapy is needed.

* Pathogenesis:

  - line 97: please add also ref Finalet Ferreiro J, et al. Am J Transplant 2016;16:414-25.

- line 98-100: do the authors mean EBV negative versus EBV positive? Now it's not clear what is the difference between EBV-negative cases (which are mostly monomorphic) and monomorphic PTLD presenating as DLBCL?

- line 108-118: the authors should mention the 9p24.1 gain, leading to PDL2 overexpression in EBV positive cases (Finalet Ferreiro J, et al. Am J Transplant 2016;16:414-25).

3. Pathologic classification:

- line 162: the authors should aso mention CLL as not considered PTLD, but also (EBV-associated) MALT lymphoma, which is considered PTLD.

 - Table 1: typo : early insted of ealry (2x).

4. Clinical features:

- line 211: 'response to immunotherapy reduction' should be 'respons to reduction of immune suppressive treatment".

- line 215-216: a reference should be added.

6. Risk factors:

- table 2: post-SOT weak evidence: 1 (type of graft) and 4 can should be combined. The word "cadaveric" is not encouraged, please use "graft from a deceased donor". 'Spleenectomy' should be 'splenectomy', 'Dgree' should be 'degree'.

7. Surveillance and prevention:

- line 291: is a preemptive therapy (as discussed in 8. premptive therapy).

8. Preemptive therapy

- line 308: I would suggest to delete 'when EBV load surpasses 1000...) as many centers use other strategies (f.e. 1 log increase in 1 or 2 weeks,....).

9. Treatment

- line 390: recently a long term follow up of this study was published (Gonzalez-Barca E, et al. Ann Hematol 2021;100:1023-9).

- line 397-399: is this correct? In PTLD-1 trial there was no difference between EBV+ and EBV- cases. Please comment.

- Table3: references should be added. 

- Table 3: ALLELE study: first outcomes have been published (as discussed in the text - Prockpop S, et al. J Clin Invest 2020;130:733-47).

- Line 481: ORR of 80% and 100%. Are these numbers correct?? If you combine CR and sustained PR response rates were 68% (HSCT) and 54% (SOT). What do the authors mean with respectively (SOT vs HSCT?) . This should be clarified.

- line 498-505: the authors describe 2 case series, but there are no references.

- line 521: 'ibrtumomab' should be 'ibritumomab'.

- line 525-537: any experience with BV in classic HL-type PTLD?

- line 571: delete 'is'.

- line 638-641: the statement on 'authors concluded' refers to ref 120, but this should be ref 121

- line 632: 7-10 post-kidney : do you mean years?

-  CART: the authors should also include the case series with tisa-cel (Luttwak E, eat. Bone Marrow Transplant 2021;156;1031-7)

 - line 666: two rare subtypes should be briefly added: Classic Hodgkin lymphoma and plamasblastic lymphoma-PTLD

- 698-710: ref 133 and 134 only includes pediatric PTLD. As the artice deals with adult PTLD and no other pediatric trials are mentioned, this can be omitted.

- 698-710: a short summary of the discussion on treatment of BL should be added (Dierickx D, et al. Blood 2015;126:2274-83) .

References:

please check double citations (f.i. ref 24/50 and 13/70 are identical).

Congratulations with this nice work,

Author Response

Please see comments adressed at the pdf document attached.